# Injury Occurrence in Amateur Rugby: Prospective Analysis of Specific Predictors over One Half-Season

**DOI:** 10.3390/medicina59030579

**Published:** 2023-03-15

**Authors:** Luka Bjelanovic, Dragan Mijatovic, Damir Sekulic, Toni Modric, Marijana Geets Kesic, Aleksandar Klasnja, Patrik Drid, Sime Versic

**Affiliations:** 1Rugby Club Nada, 21000 Split, Croatia; 2Faculty of Kinesiology, University of Split, 21000 Split, Croatia; dado@kifst.hr (D.S.);; 3Faculty of Health Sciences, University of Mostar, 68000 Mostar, Bosnia and Herzegovina; mijatovicdragan@ymail.com; 4High Performance Sport Center, Croatian Olympic Committee, 10000 Zagreb, Croatia; 5Faculty of Medicine, University of Novi Sad, 21000 Novi Sad, Serbia; 6Faculty of Sport and Physical Education, University of Novi Sad, 21000 Novi Sad, Serbia

**Keywords:** rugby union, soft tissues, bones, playing positions, pain, injury prevalence

## Abstract

*Background and objectives:* The incidence of injuries in rugby is extremely high, but studies have rarely examined the predictors of injury in amateur players. This study aimed to systematically analyse sports and injury factors as potential predictors of musculoskeletal injuries in senior-level amateur rugby players. *Methods*: The participants in this study were 101 senior-level rugby players from Croatia (average of 24.64 years old). At baseline, all participants were tested on sociodemographic and anthropometric parameters (age, body height and mass), consumption of dietary supplements, preseason injury status and training volume, and sport factors (position in game). Data on injury occurrence (dependent variable), prevalence of pain, training status, and characteristics of the played match were surveyed prospectively once a week during the three-month period (one half-season). *Results*: The logistic regression revealed a higher injury occurrence in forward players of the 1st row, 2nd row (OR = 5.07; 95% CI: 1.64–15.69), and center (OR = 4.72; 95% CI: 1.28–14.31), with reference to outside back players. When observed univariately, higher body mass, higher level of competition, more weekly training sessions, self-perceived pain, and playing with pain were significant injury risk factors. The multivariate logistic regression identified pre-season injury (OR = 1.30, 95% CI: 1.09–1.52), higher level of the game/match (OR = 1.44, 95% CI: 1.13–1.76), higher body mass (OR = 1.03, 95% CI: 1.01–1.05), and pain prevalence (OR = 5.71, 95% CI: 3.22-7.70) as multivariate predictors of injury occurrence over the season. *Conclusion*: The results of this study showed that among sport factors, the playing position, level of competition, and training exposure represent major injury risk factors. Therefore, in order to reduce the number of injuries, special emphasis should be placed on the specific tackling technique of forward players, which could both increase their situational efficiency and protect them from injuries. Additionally, perceived pain, injury history, and playing with injury were noted among injury factors as the ones that can be predictors of future injuries. In that manner, it is important that coaching and medical staff monitor players with previous injuries and with pain symptoms in order to act preventively against injury occurrence.

## 1. Introduction

Rugby union, or simply rugby, is a contact team sport characterized by dynamic actions consisting of fighting for ball possession and advancing a team’s position in the field [1]. The main task in rugby is to bring the ball in the opponent’s goal space (in-goal area). A rugby game has two 40 min periods, during which physical collisions and duels with the opponents are constantly performed, along with individual and team technical–tactical tasks. It is a globally popular sport, with more than 120 countries in the governing World Rugby federation and 9 million players worldwide [2]. Teams consist of 15 players who are classified in 2 main groups according to their positions—backs and forwards. Additionally, forwards are divided in front, second, and back rows, while backs include halves and inside and outside backs [3].

Due to its contact nature and frequent tackles and collisions, injury incidence in rugby is significantly higher than in other sports [1]. In particular, studies reported around 80 injuries per 1000 player match hours in professional rugby clubs [4,5]. This rate is lower in amateur rugby, with the incidence ranging from 5.95/1000 to 59.2/1000 player match hours [6]. The large differences among these findings can be explained by the competition level, players’ quality, and other environmental factors, such as surfaces and equipment [2]. The most common types of injuries in rugby are sprains and ligament injuries, while the most common injury locations are the head and face [1]. On the other hand, the location with the highest overall burden is the knee (11.1 days/1000 h) [1].

Studies reported that 61% of all rugby injuries occur as a consequence of physical contact among players during tackling in professional rugby, while this percentage is lower (47.9%) in amateur rugby [7,8]. Tackling is the most frequent contact situation during rugby, and it occurs on average 221 times in a professional rugby game [7]. With the development of rugby over the years, players have become stronger and faster; moreover, there is a noticeable trend of selection of larger and more massive individuals [9]. It is clear that all these factors influence the generation of more significant forces in the contact among players.

Efforts of sport scientists and practitioners to prevent injuries and reduce their severity have led to an increase in research of potential risks or protective factors for injury occurrence [10,11,12]. Studies of various sports have analysed different groups of factors, such as the individual psychosocial and biological characteristics of athletes, the levels of motor and functional abilities, and environmental and sport factors [12,13,14]. Injury investigation in rugby most commonly occurs in descriptive studies that analyse injury occurrence and distribution by type, body part, etc. [15]. However, recently, a number of studies have explored the potential predictors of injuries in rugby [16,17,18,19,20].

For example, Fuller, Brooks, Cancea, Hall and Kemp [20] investigated the types of contact and their associations with injuries in the context of English Premiership. Their most significant finding was that tackles were responsible for the highest number of injuries and the greatest loss of time in matches (five times more than any other contact) [20]. A study of professional rugby players from English Premiership teams found that the training load had a significant influence on injury occurrence, as players had an increased risk of injury if they had high one-week cumulative loads or large week-to-week changes in the total load [17]. Similar to this, the match load was also found to be a significant risk factor by a 7-season prospective cohort study; players who had participated in less than 15 or more than 35 matches over the preceding 12-month period were found to be more prone to injury [19].

Regarding specific motor abilities, a few studies analysed the mobility and stability of rugby players with respect to fundamental movement patterns [16,18,21]. All these studies indicated low scores on functional movement screen (FMS) tests as risk factors for injuries in rugby players [16,18,21]. Finally, one of the factors that was evidenced as an important injury predictor is the playing position of forward players, who were generally found to present higher injury occurrence than backs [22,23,24].

Rugby is a contact sport played at the amateur, semi-professional, and professional level, with a significant number of players on the amateur level, i.e., those who are not paid for training and competition, but play rugby alongside their primary jobs. While there is a significant number of studies of injuries in professional rugby, a limited body of knowledge exists with respect to amateur, non-professional rugby [6]. This is especially evident when considering studies that investigate potential groups of factors as predictors of injuries. It is clear that injuries represent serious health problems in rugby; therefore, they should be analysed in detail. The main aim of this study was to systematically investigate groups of sports and injury factors as potential predictors of injuries. We hypothesized that playing position, training volume and injury history will be related to injury occurrence.

## 2. Materials and Methods

### 2.1. Participants and Design of the Study

The sample of participants in this study initially consisted of 122 senior rugby players from Croatia who were members of all first league senior clubs in Croatia. From the final analysis, we excluded all players who did not regularly complete the weekly surveys. The final sample comprised 101 participants (average of 24.64 years old). Since the study comprised practically the whole population of amateur rugby players from Croatia, the a priori calculation of necessary sample size was not done. The participants were not rugby professionals, as the majority were students or had other primary jobs, with 75.91% of them with more than 40 h of hard physical work per week. They had on average 14.76 years’ experience in sports and 9.11 years in rugby in particular. Most of them (44.04%) trained 3 or more times a week, 31.49% trained twice a week, 10.54% once, and 13.93% had no training per week.

This prospective study was conducted during the second competitive half-season of the Croatian first national rugby division, which lasted three months. At the study baseline, we collected independent variables, while injury occurrence (dependent variable) was collected during the competitive season. The study design is presented in Figure 1.

Besides the Croatian league (lowest competitive level of the game), in the same period, teams were also included in the Regional Rugby Championship along with the teams from Slovenia and Bosnia and Herzegovina (higher level of the game), while some players participated in national team games (highest level of the game). During the observed period, players participated in a maximum of one friendly game, three games of the Croatian League, three games of the Regional Championship, and two national team games. Baseline testing was performed at the beginning of the season, when all participants filled in the designed online questionnaire. During the season, every week after the matches, the examiners contacted the participants and reminded them to fill in the online questionnaire in order to record any new injuries that had occurred in the past week during training or matches. The weekly responses were later observed as study entities. Injury was defined as a condition of a part of the body that has prevented them from participating in normal playing or training in the next match or in training that was to take place at least 24 h after the moment of occurrence of injury. All testing was conducted in agreement with the national rugby federation and the clubs. The study was conducted according to the guidelines of the Declaration of Helsinki and was approved by the Institutional Ethics Committee of the Faculty of Kinesiology, University of Split.

### 2.2. Variables

Baseline testing allowed us to collect all data of relevant independent variables by direct measurement (see later for details) and with our designed survey questionnaire, which was constructed based on questionnaires previously applied in injury research on rugby players, football referees, Norwegian Olympians and Paralympians from multiple sports, and handball and tennis players [20,25,26,27,28]. The participants were informed that by completing the questionnaire, they give their consent that this and future data can be collected for the research and that their personal data will remain anonymous and be used only for contact during the research.

The independent variables included age, experience in rugby, body height, body mass, calculated body mass index (BMI), history of injuries (injuries in the last 12 months), sport factors, musculoskeletal problems and pain during training and matches, and consumption of dietary supplements. Age and experience in rugby were evidenced in years. Body height and mass were recorded by medical examination at the beginning of the season with usage of the Seca stadiometers and scales (Seca, Birmingham, UK), and BMI was calculated (BMI = mass (kg)/height (m)^2^). Dietary supplementation was evidenced on a binomial (yes–no) scale. Apart from experience in rugby sport (see previously), the sport factors included: playing position (first row, second row, back row, half backs, centres, outside backs), number of trainings during preseason, and number of trainings per week (during the season). Finally, occurrence of pain was also recorded on a binomial (yes–no) scale.

All injury data (dependent variables) were collected according to the consensus on rugby injuries established by Fuller et al. [29]. Accordingly, injury was defined as any physical complaint caused by a transfer of energy that exceeded the body’s ability to maintain its structural and/or functional integrity, which was sustained by a player during a rugby match or rugby training, irrespective of the need for medical attention or time lost from rugby activities [29]. Medical-attention injuries were excluded if the player was fit for training or a match the day after. Injuries were recorded according to their topological location on the body, the type of injury, the side of the body affected, and the event preceding the injury. They were also classified according to the event during which they occurred (match or training) and according to whether there had been contact with another player (or object) or not (i.e., overuse injuries). For injuries that had occurred during contact, activities were recorded depending on whether the player had been the tackler or had been tackled and whether it was a scrum, ruck, or maul; a collision; or something else.

### 2.3. Statistics

The normality of the distributions was checked for all variables with the Kolmogorov–Smirnov test. Consequently, the descriptive statistics for parametric variables included the calculation of means and standard deviations, and for nonparametric variables, frequencies (counts) and percentages.

The analysis of the associations between independent and dependent variables was done throughout univariate and multivariate logistic regression analyses. In the first phase, all independent variables were univariately correlated to injury occurrence. In the second phase, all significantly associated predictors were simultaneously included in the multivariate logistic analysis in order to control possible confounding effects. The odds ratio (OR) and 95% confidence interval (95% CI) were reported. The Hosmer–Lemeshow test (HL) was used to evaluate the appropriateness of the model’s fit (with a significant chi-square value indicating an inappropriate model fit).

Statistica version 13.5 (Tibco Inc., Palo Alto, CA, USA) was used for all analyses, with a *p*-level of 0.05.

## 3. Results

The studied players reported 54.11 injuries (95% CI: 39.88–70.11) per 1000 h of game, and 5.29 injuries (95% CI: 3.11–7.34) per 1000 h of training. In total, 37 injuries (77.1%) were traumatic/acute, while 11 injuries (22.9%) were evidenced as being chronic in nature (overuse injuries). Table 1. presents the number of days each player was sidelined after injury occurrence, with most of them out between 1 and 4 weeks, while only 6 players were out for more than 28 days.

Of the chronic/overuse injuries, ligament injuries were most common (14 injuries in total, 30.4% of all injuries reported), followed by hematoma/contusion/bruise (12, 26.1%) and tendon injuries (6, 13%). Serious traumatic injuries were evidenced as follows: brain concussion (3, 6.52%), fracture (2 injuries, 4.4%), and dislocation/subluxation (2, 4.35%). Other types of injury (muscle rupture, nerve injuries, ankle sprain, etc.) were less common (each <2% of all injuries).

As it is shown in the Table 2. the knee was the most common injured body location (9 injuries, 18.8% of all injuries), followed by the shoulder/clavicle (8 injuries, 16.7%), ankle (5 injuries, 10.4%), head (4 injuries, 8.3%), and chest/ribs and lower back (3 injuries each, 6.3% each).

Injuries mostly occurred during tackling (21 injury, 46%), followed by collision and scrum (4 injuries each, 4.81% each) and ruck (3 injuries, 8.11% of all occasions).

The univariate associations between anthropometric and sociodemographic variables in relation to injury occurrence are presented in Figure 2. Body mass was found to be significantly associated with injury occurrence (OR = 1.02, 95% CI: 1.01–1.04), with a higher risk for injury in heavier players.

Body mass index (kg/m^2^)

**Figure 2 medicina-59-00579-f002:**
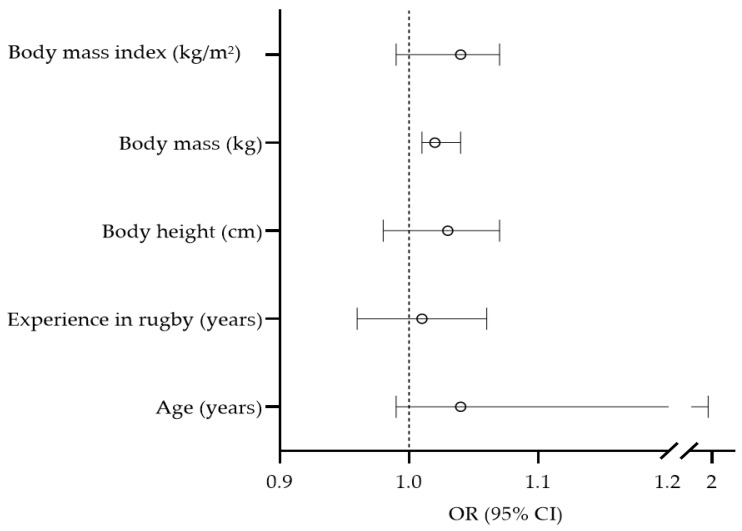
Univariate associations between sociodemographic and anthropometric variables with injury occurrence.

The association between the position played in a rugby game/match and injury is presented in Figure 3A, where five playing positions (1st row, 2nd row, 3rd row players, Half backs, and Center) are related to the injury occurrence of outside backs (as a reference value). Specifically, higher risk was evidenced in 1st line players (OR = 3.57, 95% CI: 1.12–11.35), 2nd line players (OR = 5.07, 95% CI: 1.64–15.62), and centres (OR = 4.72, 95% CI: 1.28–14.31) compared to that of players playing in the last line. When correlating the number of training sessions the players participated in over the week before the injury occurrence with the occurrence of injury (Figure 3B), those players who did not participate in training had a lower risk of being injured than those who participated in three training sessions or more during the week preceding the injury (OR = 0.38, 95% CI: 0.11–0.60).

The last set of univariate logistic regressions included specific factors related to injury occurrence. The risk for injury over the competitive period was higher in players who suffered injury in the preseason (OR = 1.43, 95% CI: 1.21–1.69). Further, players who reported the consumption of dietary supplements at the study baseline were at higher risk for being injured during the competitive period (OR = 1.30, 95% CI: 1.06–1.60). Finally, the risk for injury occurrence was higher in players who reported the consumption of dietary supplements at the study baseline (OR = 1.59, 95% CI: 1.29–1.96) (Figure 4).

The last phase of the analyses of the association between the studied variables and injury occurrence comprised multivariate logistic regression, where predictors previously found to be significantly associated to injury occurrence were simultaneously included in the analysis in order to control for possible covariates. However, in this analysis, we intentionally did not include the playing position and number of training sessions per week as potential predictors due to their multinomial nature (please see Figure 2 for details). The multivariate logistic regression identified pre-season injury (OR = 1.30, 95% CI: 1.09–1.52), higher level of the game/match (OR = 1.44, 95% CI: 1.13–1.76), higher body mass (OR = 1.03, 95% CI: 1.01–1.05), and pain prevalence (OR = 5.71, 95% CI: 3.22-7.70) as multivariate predictors of injury occurrence over the season (Figure 5). The HL test indicated an appropriate model fit (Chi square = 4.11, *p* > 0.05).

## 4. Discussion

This study aimed to detect injury risk factors among amateur rugby players. Several important findings are reported here. First of all, injuries most often occurred at the front and second row and centre positions, and injured players were heavier than those players who were not injured. The level of competition was found to be a predictor of injuries, with more injuries being detected at higher competitive levels. Furthermore, the risk of getting injured was lower among players who did not participate in training during the week and only played games. Finally, a player’s injury status was associated with injury occurrence, with (i) perceived pain, (ii) injury history, and (iii) playing while injured all being identified as risk factors. In general, our initial study hypothesis was confirmed.

### 4.1. Playing Position and Body Mass

As mentioned in the introduction, rugby is characterized by a significant number of contacts, such as collisions, tackles, and pulls. There are no doubts that these playing situations most frequently involve forwards. Particularly, studies reported that of almost 400 contacts occurring per game, forwards are involved in more than 2/3 [30,31]. Such large numbers of contacts occurring in games and training sessions present a greater possibility to sustain injury when two or more bodies of a certain mass and speed collide [32]. Supportively, two studies revealed an association between playing position and injury prevalence [8,33]. In particular, a study of amateur New Zealand rugby players revealed that forwards sustained 56.1% of total injuries, while 43.9% were sustained by back players [8]. Similarly, a study of Argentinian players highlighted playing positions as injury predictors, with forwards being at a higher risk; especially, flankers sustained 15.5% of all recorded injuries [33]. Therefore, our results are generally in agreement with previous reports in which the authors confirmed playing positions as factors of influence on injury occurrence in rugby.

The emergence of body mass as a risk factor for injury should also be explained in the context of positional specificities in rugby. As mentioned, forward players participate in a large number of several types of contact situations. In all these contacts, the amount of force production is, along with proper technique, the key factor for situational efficiency. It is well known that a higher body mass generates more force if moving at the same speed [34]. Since body mass is a more trainable force-determining factor than speed (i.e., it is easier to increase a players’ body mass than to improve their speed performance), it is clear that forward players benefit from a greater body mass, since it allows them to more effectively overcome the opponent [35]. Supportively, body mass was recognized as a statistically significant determinant of success in professional rugby [36]. Interestingly, the data reported in the same study directly support our previous consideration on body mass as a factor of success, since they show a trend of increased body mass (6.63 and 6.68 for back and forward players, respectively) in the period between 1987 and 2007 [36]. Additionally, a study that covered rugby players considering the whole 20th century found an increase in body mass of 2.6 kg per decade, which was far above the increase in the general population of young men [37]. Although previous studies have dealt with professional players, and while the participants in our study were amateur players, it is clear that body mass influenced their performance in the same manner. As a result, it is reasonable to assume that forwards, besides the fact that they are more frequently in risky situations, have higher possibilities of getting injured, as they use a greater amount of force in collision situations because of their body mass.

### 4.2. Training and Competition

The competition level was found to be a factor influencing injury occurrence, with higher injury occurrence during games at higher competitive levels. Results from previous studies support this finding. For example, a study of English amateur players noted that the number of injuries increased with the increase in the level of competition (level A, 21.7; level B, 16.5; and level C, 14.2 injuries per 1000 match hours per player) [38]. This difference is also present between the amateur and professional levels, as epidemiological studies reported more injuries among professionals [4,5,39]. Specifically, studies of professional players noted values between 80 and 90 injuries per 1000 h per player [4,5,39]. On the other hand, this number was significantly lower in amateur players, and it dropped below 50 and, in some studies, even below 20 injuries per 1000 h [6,38].

There are two possible explanations for this finding. The first one is related to the quality of the opponent, as a higher level of competition implies players of higher quality. The difference in functional and motor abilities among players of different quality levels was found at both the junior and senior levels [9,40]. At higher performance levels, players are faster, stronger, and more massive, which undoubtedly affects their execution, allowing them to conduct a faster game and more forceful direct collisions, which altogether increases the risk of injury. Second, it is important to note that a higher level of competition implies a longer time span of having the ball in play or a longer “clean play”; a study noted that the amount of time the ball is in play increases with the level of competition quality [41]. As the playing time increases, the chance of being injured is more pronounced, as a larger number of risky contacts occur.

Although controversial at first sight, finding a negative correlation between injury occurrence and training hours is logical. In fact, our results suggest that players attending no training sessions during the week are less likely to be injured than their colleagues who train three or more times per week. Specifically, these players have a significantly lower time of exposure to specific rugby demands and, in that way, are at a lower risk of getting injured. However, it is important to point out that a reduced number of training hours is not desirable and will not always lead to a reduced number of injuries. Moreover, reduced training will cause reduced levels of strength, power, and endurance. However, considering the contact nature of the sport and the amateur level of competition, it is obvious that in this observed sample, the reduced exposure to specific efforts to some extent outweighs the potential training benefits.

### 4.3. Injury and Pain Status

One of the main predictors of injury occurrence in this study was the history of injuries. Interestingly, in this regard, previous studies offered inconsistent results. For instance, New Zealand researchers did not obtain a significant association between pre-season injuries and injuries during the season in rugby players [42]. However, in the same study, there was a correlation between injuries suffered in the previous season and the ones in the current season [42]. On the other hand, some studies from other similar sports revealed similar findings, such as those reported here [43]. Specifically, a study of Australian football players found significant associations between pre-season injuries and injuries that occurred during the competition period [43].

There are several possible explanations for these findings. First of all, the reduced training intensity during the pre-season (due to injury) and the sudden increase in intensity at the beginning of the competition period can affect the occurrence of new injuries. Supportively, several studies reported that a sudden increase in the training load can cause a number of injuries in players [17,44]. Additionally, a study of English professional rugby players detected a higher risk of injuries if players increased their weekly load [17]. Second, insufficient and inadequate rehabilitation and recovery can cause future problems with re-injuries [17]. As a result, it is clear that special attention should be paid to the load management of players who return to practicing sports after a specific injury.

The risk of injury was high among players who reported pain in the period that preceded the injury, but not necessarily in the body part that eventually got injured. Although the pain did not specifically relate to the injured part of the body, it can be assumed that it still affected the performance of the players. Consequently, pain causes an elongation of reaction time, impairs motor control, and can increase inaccuracy when performing specific game tasks [45]. In addition, players who feel pain try to spare the affected body part, which leads to compensational movement, the overload of some other tissue, and, possibly, injuries [46,47,48].

### 4.4. Strengths and Limitations of the Study

One of the limitations of this study was the fact that the authors could not be absolutely certain about the accuracy of the responses, and it is possible that some injuries and feelings of pain remained unreported. In addition, no details about external factors at each training session and match (weather conditions, hardness of the field) were recorded, so there is space for improvement in future studies. The survey did not check the biomechanical abnormalities, muscular imbalances, and other risk factors related to motoric abilities, and this should be investigated in future studies. Further, no data regarding individual nutrition and supplement consumption were recorded, which could affect player’s injury status. An additional limitation is related to anthropometry measures, as we did not include skinfold measurements and body fat mass, which could provide more explanation regarding the association between body mass and injuries. In future studies, the body composition of the rugby players should be analysed in detail. Finally, the injuries were self-reported and, in some cases, were not confirmed by professional medical staff. However, the authors regularly contacted all participants after each match and interviewed them in detail, thus properly collecting all the necessary data. Regarding these limitations, in future studies, variables about external factors and nutrition habits should be included, along with more specific details regarding occurred injuries, which should be collected by a medical specialist.

The main strength of this study was the sample of participants, as it included the entire population of amateur rugby players from all clubs at the highest Croatian national competitive level. In addition, in the process of data collection, the online validated questionnaire was used, which is a novelty in the surveying of amateur rugby players, and it included a large number of parameters. Therefore, considering the lack of studies of injuries in amateur rugby players, especially regarding injury predictors, the authors firmly believe that the findings of this study can help improve the general health status of amateur rugby players.

## 5. Conclusions

The results of this study showed that among sport factors, the playing position, level of competition, and training exposure represent major injury risk factors. In particular, forward players and centre players were at a higher risk of getting injured than their peers playing in other positions in rugby games. This could be related to their specific position-related tasks and the fact that they were involved in many more tackles. In this context, the association between higher body mass and the increased risk of injury can also be interpreted as forward players being more massive than backs. In the future, the playing demands are unlikely to change, and the number of contacts is likely to at least remain at the same level, so it is clear that certain improvement should be looked at in the technique of specific rugby tackling and grappling. Therefore, our results provide practical implications that can be used as prevention strategies for forward players. In detail, coaches should include the specific training/mastering of the specific tackling technique in order to optimize players’ movement. This could both (i) positively influence their effectiveness in these sport situations and (ii) have a positive impact on the players’ safety.

Regarding injury factors as potential predictors of injuries, our results highlighted perceived pain, injury history, and playing with injury, as players who suffered injuries in the pre-season period and/or reported pain before training/games were also at a high risk of being injured. Coaching and medical staff should pay special attention to those players both in the training process and during the selection of the squad for a game. It is well known that players often try to hide symptoms and problems because of their desire to play and not be excluded from the squad. However, staff members in day-by-day club work need to focus on player’s wellbeing, pain, and injury status. In particular, they have the task to have regular and honest communication with players, which would allow them to identify potential problems and to act preventively against injury occurrence.

In future studies, a wider spectrum of predictors should be included (different motor abilities, body composition, muscle imbalances, and biomechanical abnormalities) in order to create a clearer picture of injury risk in amateur rugby.

## Figures and Tables

**Figure 1 medicina-59-00579-f001:**
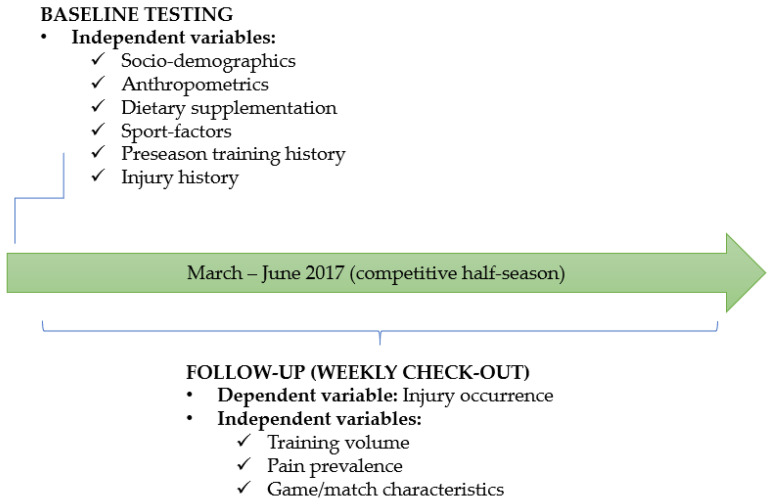
Study design.

**Figure 3 medicina-59-00579-f003:**
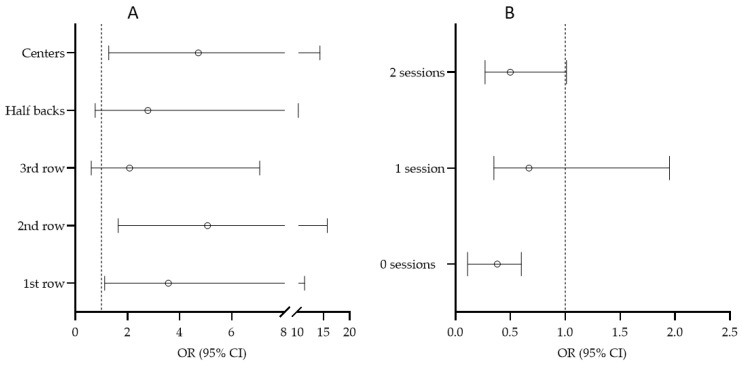
Univariate associations between (**A**) playing positions (with players playing in the last line (Outside backs) as reference value) and (**B**) number of training sessions in a week before the match (with three sessions or more per week as reference value) with injury occurrence.

**Figure 4 medicina-59-00579-f004:**
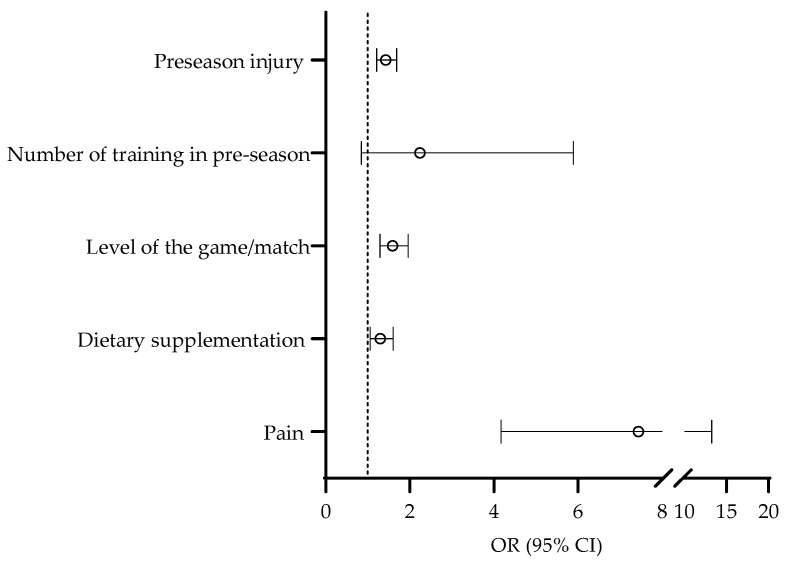
Univariate associations between preseason injury occurrence, preseason training, level of the game played, dietary supplementation, and pain prevalence with injury occurrence.

**Figure 5 medicina-59-00579-f005:**
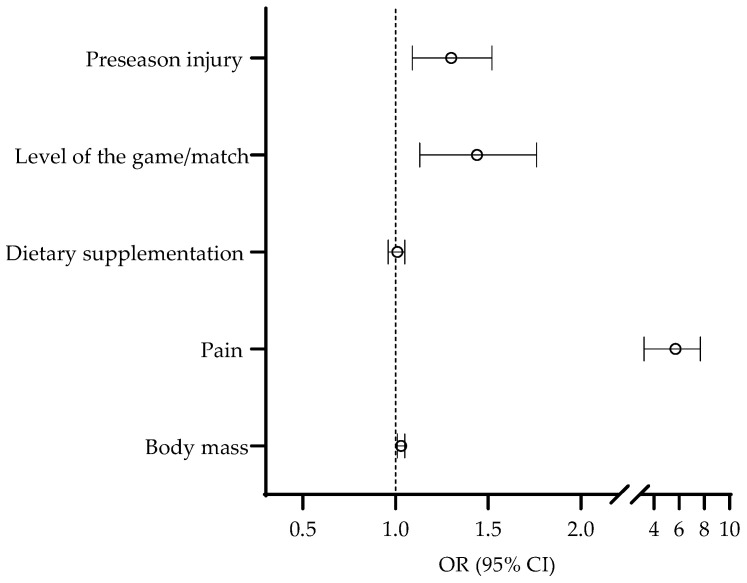
Multivariate associations between preseason injury occurrence, level of the game played, dietary supplementation, pain prevalence, and body mass with injury occurrence.

**Table 1 medicina-59-00579-t001:** Period out of sport practice for injured players.

	All		Forwards		Scrum	
	*n*	%	*n*	%	*n*	%
0–1 days	4	9.30	2	12.50	2	7.41
2–3 days	9	20.93	3	18.75	6	22.22
4–7 days	14	32.56	4	25.00	10	37.04
8–28 days	10	23.26	6	37.50	4	14.81
More than 28 days	6	13.95	1	6.25	5	18.52

**Table 2 medicina-59-00579-t002:** Anatomical location of injuries.

	All		Forwards		Scrum	
	*n*	%	*n*	%	*n*	%
head	4	8.3	3	17.6	1	3.2
neck	1	2.1	0	0.0	1	3.2
shoulder/clavicle	8	16.7	1	5.9	7	22.6
chest/ribs	3	6.3	2	11.8	1	3.2
upper back	1	2.1	0	0.0	1	3.2
lower back	3	6.3	0	0.0	3	9.7
forearm	1	2.1	0	0.0	1	3.2
elbow/	1	2.1	0	0.0	1	3.2
hand/finger/thumb	2	4.2	1	5.9	1	3.2
wrist	2	4.2	0	0.0	2	6.5
hip/groin	1	2.1	1	5.9	0	0.0
front part of upper leg	1	2.1	0	0.0	1	3.2
back of thigh	1	2.1	1	5.9	0	0.0
knee	9	18.8	4	23.5	5	16.1
lower leg	1	2.1	0	0.0	1	3.2
ankle joint	5	10.4	2	11.8	3	9.7
foot/toes	1	2.1	1	5.9	0	0.0

## Data Availability

The data are available upon reasonable request.

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
