# Peer review of "Injury Occurrence in Amateur Rugby: Prospective Analysis of Specific Predictors over One Half-Season"

_medicina, 2023, doi:10.3390/medicina59030579_

Round 1
Reviewer 1 Report
Dear authors,
About my notes in the article
Abstract: The conclusion of the abstract does not respond to the proposed objective of the study. The objective of this study was not to evaluate which strategies can be adopted to reduce the risk of injurie.
Introduction
The rationality of the study is unclear, what is the exact gap in the knowledge that your study hopes to fill?
What criteria used to classify athletes as amateurs?
Method
Did the BMI assessment in the athlete follow any special scale? Why has skinfold evaluation not been performed? For evaluation of body mass of athletes it is important to follow International society of sports nutrition position stand
Results
The presentation of results in table 2 is unclear, the anatomical regions presented are not in anatomical order, for example, the presentation of the result for shoulder and below for ankle appears. The presentation criterion also does not follow a criterion of higher to the lowest percentage. This impairs the interpretation of the results.
Regarding the results presented in Figure 2, I am concerned that body mass is directly correlated with the risk of injury because the percentage of fat in this analysis was not considered. Because the interpretation is that athletes with the same body weight but with different body compositions have the same risk of injury?
What were the supplements ingested by the athletes?
Discussion
The article mentions the relationship between body mass and the number of injuries, however body mass is not recommended for athletes according to the International society of sports nutrition.
Conclusions
Are not responding to the objective. Needs to be reviewed to meet the study objective
Author Response
Dear authors,
About my notes in the article
Abstract: The conclusion of the abstract does not respond to the proposed objective of the study. The objective of this study was not to evaluate which strategies can be adopted to reduce the risk of injurie.
Response: Thank You for your suggestion, we amended the conclusion part in the abstract and it now reads:
“Results of this study showed that among sport factors, playing position, level of competition and training exposure represent major injury risk factors. Therefore, in order to reduce the number of injuries, special emphasis should be placed on the specific tackling technique of forward players, which could both increase their situational efficiency and protect them from injuries. Additionally, perceived pain, injury history and playing with injury were noted among injury factors as the ones that can be predictors of future injuries. In that manner, it is important that coaching and medical staff monitor players with previous injuries and with pain symptoms in order to act preventively against injury occurrence.”
Introduction
The rationality of the study is unclear, what is the exact gap in the knowledge that your study hopes to fill?
Response: Thank You for your comment. The main rationale for this study was the knowledge gap in the field of injuries in amateur rugby. Rugby is a contact sport played at amateur, semi-professional and professional level with significant number of player on the amateur level and the main aim was to investigate injuries in this population with special emphasize on injury predictors. We added the sentence in this part in the introduction chapter and it reads:
“Rugby is a contact sport played at amateur, semi-professional and professional level with significant number of player on the amateur level, i.e. those who are not paid for training and competition but play rugby alongside their primary jobs. While there is a significant number of studies of injuries in professional rugby, a limited body of knowledge exists with respect to amateur, non-professional rugby. This is especially evident when considering studies that investigate potential groups of factors as predictors of injuries”
What criteria used to classify athletes as amateurs?
Response: Thank You, we characterized the observed players as amateurs as they are not paid or have contracts with the clubs, but play rugby alongside their primary jobs. We added the sentence that explains this (please see the previous comment, and our answer)
Method
Did the BMI assessment in the athlete follow any special scale? Why has skinfold evaluation not been performed? For evaluation of body mass of athletes it is important to follow International society of sports nutrition position stand.
Response: Thank You for your comment. We asses only body height and body mass measures with usage of the Seca stadiometers and scales (Seca, Birmingham, UK) and calculated BMI with BMI = mass (kg)/ height (m)2 formula. We realize that information about body fat mass percentage would be valuable and we added this in the limitation paragraph.
“Additional limitation is related to anthropometry measures, as we did not included skin-fold measurements and body fat mass which could provide more explanation regarding association between body mass and injuries. In the future study body composition of the rugby players should be analyzed in detail”
Results
The presentation of results in table 2 is unclear, the anatomical regions presented are not in anatomical order, for example, the presentation of the result for shoulder and below for ankle appears. The presentation criterion also does not follow a criterion of higher to the lowest percentage. This impairs the interpretation of the results.
Response: Thank You for your comment, we amended the table and the body regions are now presented in the anatomical order.
Regarding the results presented in Figure 2, I am concerned that body mass is directly correlated with the risk of injury because the percentage of fat in this analysis was not considered. Because the interpretation is that athletes with the same body weight but with different body compositions have the same risk of injury?
Response: Thank You for your comment. We understand that the absence of the body composition measures represents specific limitation and we added this in the limitation paragraph. For sure this information would provide us more detail, but considering the population we observe, we assumed that body mass is strongly related to playing position as players in amateur rugby are selected, among other, according to that criterion. So, we explained the emergence of body mass as a risk factor for injury in the context of positional specificities in rugby.
What were the supplements ingested by the athletes?
Response: Thank You. In the questionnaire we offered following options for the question about the supplements:
- No
- Yes, vitamins and minerals
- Yes, isotonic drinks
- Yes, other supplements (proteins, carbs, etc…)
We did not analyse the results for specific answer, but for usage of the supplementation in general.
Discussion
The article mentions the relationship between body mass and the number of injuries, however body mass is not recommended for athletes according to the International society of sports nutrition.
Response: Thank You for your comment. We realize this is study limitation and we added this in the study limitation paragraph. As You comment this issue in some of the previous comments, please see the answers there as they refer also to this comment.
Conclusions
Are not responding to the objective. Needs to be reviewed to meet the study objective
Response: Thank You. As I mentioned in one of the previous comments, we amended the conclusion part in both abstract and conclusion paragraph to be more related to the study objective. Please see the conclusion paragraph for more details.
Reviewer 2 Report
1. I have serious concerns about the population selected. In the title and objective, it is amateur, and in the methodology, they are senior rugby players who are members of the first league senior clubs. Lines 118 -122 also have a contradictory statement
2. Only 122 rugby players in the whole of Croatia!
3. Whether participants were screened for any prior injury before the commencement of the study
4. What about biomechanical abnormalities, muscle strength imbalances and other intrinsic factors that may risk injuries? Were these factors screened prior to participation? Are they included in the weekly survey? These are the major risk factors and predictors of injury. Without this information, a survey on predictors of injury is incomplete and biased.
5. Another major issue with the study is any medical professionals do not verify the injuries. In addition, overuse injuries are usually not reported.
6. The conclusion in the abstract is not in line with the objective of the study
7. The discussion should include each finding of the study
Author Response
- I have serious concerns about the population selected. In the title and objective, it is amateur, and in the methodology, they are senior rugby players who are members of the first league senior clubs. Lines 118-122 also have a contradictory statement
Response: Thank You for your comment. Although it sounds contradictory, it is not, as members of the first league senior clubs are all amateurs, without contracts and have primary jobs alongside with rugby.
- Only 122 rugby players in the whole of Croatia!
Response: Thank You for your comment. Yes, at the moment of the start of this study, there were only 5 clubs in the Croatian top division with total of 122 players.
- Whether participants were screened for any prior injury before the commencement of the study
Response: The participants were not screen by the medical professionals, but as we mentioned in the variables chapter, they filled the survey that was based on questionnaires previously applied in injury research on rugby players, football referees, Norwegian Olympians and Paralympians from multiple sports, and handball and tennis players and the questions regarding injuries in previous 12 month were included. We added short explanation in the text as it follows:
“Independent variables included age, experience in rugby, body height, body mass, calculated body mass index (BMI), history of injuries (injuries in last 12 months), sport factors, musculoskeletal problems and pain during training and matches, consumption of dietary supplements.”
- What about biomechanical abnormalities, muscle strength imbalances and other intrinsic factors that may risk injuries? Were these factors screened prior to participation? Are they included in the weekly survey? These are the major risk factors and predictors of injury. Without this information, a survey on predictors of injury is incomplete and biased.
Response: Thank You for your suggestion. We understand that injuries are multifactorial problem and that there are strong evidences that factors you mentioned can predict injuries. However, we did not included them in our research as we focused mainly on sport and injury factors. We added additional sentence in the conclusion part regarding guidelines for future studies.
“In the future studies wider specter of predictors should be included (different motor abilities, body composition, muscle imbalances, biomechanical abnormalities) in order to create more clear picture about injury risk in amateur rugby.”
- Another major issue with the study is any medical professionals do not verify the injuries. In addition, overuse injuries are usually not reported.
Response: Thank You. We are aware that this is main study limitation and we noted it in the strengths and limitations chapter: “Finally, the injuries were self-reported and, in some cases, were not confirmed by professional medical staff. However, the authors regularly contacted all participants after each match and interviewed them in detail, thus properly collecting all the necessary data. Regarding this limitations, in the future studies variables about external factors and nutrition habits should be included, along with more specific details regarding occurred injuries that should be collected by the medical specialist.”
However, we believe that with regularly contacts with the participants we collected all relevant data regarding both traumatic and overuse injuries.
- The conclusion in the abstract is not in line with the objective of the study
Response: Thank You for your comment. We amended the conclusion part in both abstract and conclusion paragraph to be more related to the study objective.
- The discussion should include each finding of the study
Response: Thank you for your suggestion. We summarized main findings in the first chapter of discussion and tried to follow them all in the following sub-chapters as it stands:
4.1. Playing position and body mass that is related to their position
4.2. Training frequency and level of the competition
4.3. Injury and pain factors
Round 2
Reviewer 1 Report
The notes were answered.
Author Response
Dear Reviewer,
Thank You for your comments. We belive that the manuscript is now improved.
Reviewer 2 Report
The paper is improved now.
However, the survey did not check the biomechanical abnormalities, muscular imbalances and other risk factors. It is a major limitation of the study.
It should be added as a limitation of the study.
Author Response
Dear Reviewer,
Thank You for your comment. We amended the limitations paragraph and added the sentence:
"The survey did not check the biomechanical abnormalities, muscular imbalances and other risk factors related to motoric abilities and this should be investigated in the future studies"